# Monocentric, Retrospective Study on Infectious Complications within One Year after Solid-Organ Transplantation at a Belgian University Hospital

**DOI:** 10.3390/microorganisms12040755

**Published:** 2024-04-09

**Authors:** Céline Van Den Daele, Delphine Martiny, Isabelle Etienne, Delphine Kemlin, Ana Roussoulières, Youri Sokolow, Desislava Germanova, Thierry Gustot, Leda Nobile, Maya Hites

**Affiliations:** 1Clinic of Infectious Diseases, Hôpital Universitaire de Bruxelles (HUB), 1070 Brussels, Belgium; celine.van.den.daele@ulb.be; 2Laboratoire Hospitalier Universitaire de Bruxelles (LHUB-ULB), Department of Microbiologie, Faculté de Médecine et Pharmacie, Université de Mons (UMONS), 7000 Mons, Belgium; delphine.martiny@lhub-ulb.be; 3Department of Pneumology, Hôpital Universitaire de Bruxelles (HUB), 1070 Brussels, Belgium; isabelle.etienne@hubruxelles.be; 4Department of Nephrology, Hôpital Universitaire de Bruxelles (HUB), 1070 Brussels, Belgium; delphine.kemlin@hubruxelles.be; 5Department of Cardiology, Hôpital Universitaire de Bruxelles (HUB), 1070 Brussels, Belgium; ana.roussoulieres@hubruxelles.be; 6Department of Thoracic Surgery, Hôpital Universitaire de Bruxelles (HUB), 1070 Brussels, Belgium; youri.sokolow@hubruxelles.be; 7Department of Digestive Surgery, Hôpital Universitaire de Bruxelles (HUB), 1070 Brussels, Belgium; desislava.germanova@hubruxelles.be; 8Department of Transplantation, Hôpital Universitaire de Bruxelles (HUB), 1070 Brussels, Belgium; thierry.gustot@hubruxelles.be; 9Department of Intensive Care Unit, Hôpital Universitaire de Bruxelles (HUB), 1070 Brussels, Belgium; leda.nobile@hubruxelles.be

**Keywords:** solid-organ transplantation, infection, immunosuppression

## Abstract

The epidemiology, diagnostic methods and management of infectious complications after solid-organ transplantation (SOT) are evolving. The aim of our study is to describe current infectious complications in the year following SOT and risk factors for their development and outcome. We conducted a retrospective study in adult SOT recipients in a Belgian university hospital between 2018 and 2019. We gathered demographic characteristics, comorbidities leading to transplantation, clinical, microbiological, surgery-specific and therapeutic data concerning infectious episodes, and survival status up to one year post-transplantation. Two-hundred-and-thirty-one SOT recipients were included (90 kidneys, 79 livers, 35 lungs, 19 hearts and 8 multiple organs). We observed 381 infections in 143 (62%) patients, due to bacteria (235 (62%)), viruses (67 (18%)), and fungi (32 (8%)). Patients presented a median of two (1–5) infections, and the first infection occurred during the first six months. Nineteen (8%) patients died, eleven (58%) due to infectious causes. Protective factors identified against developing infection were obesity [OR [IC]: 0.41 [0.19–0.89]; *p* = 0.025] and liver transplantation [OR [IC]: 0.21 [0.07–0.66]; *p* = 0.007]. Risk factors identified for developing an infection were lung transplantation [OR [IC]: 6.80 [1.17–39.36]; *p* = 0.032], CMV mismatch [OR [IC]: 3.53 [1.45–8.64]; *p* = 0.006] and neutropenia [OR [IC]: 2.87 [1.27–6.47]; *p* = 0.011]. Risk factors identified for death were inadequate cytomegalovirus prophylaxis, infection severity and absence of pneumococcal vaccination. Post-transplant infections were common. Addressing modifiable risk factors is crucial, such as pneumococcal vaccination.

## 1. Introduction

Solid-organ transplantation is the treatment of choice for end-stage organ failure [1,2]. In 2022, an estimated 157,000 organs were transplanted worldwide [3]. To prevent graft rejection, patients are treated with immunosuppressive treatments, thus increasing risk of infection [1,4]. According to recent reviews, more than 80% of solid-organ transplant (SOT) patients are likely to develop at least one infection in the first year following transplantation, causing significant morbidity and mortality [5]. In SOT patients, three periods of infections are described: early infections (from day one of transplantation to the end of the first month post-transplantation (PT)), intermediate infections (between two to six months PT) and late infections (beyond six months PT) [6,7].

To reduce these infectious risks, recipients are screened and treated for active and latent infections (e.g., tuberculosis, human immunodeficiency virus, etc.), and vaccinated against potential future pathogens [8,9]. Recommendations for vaccination against potential pathogens evolve over time; when considering *Streptococcus pneumoniae* (*S. pneumoniae*), only the 23-valent polysaccharidic vaccine was available and recommended before 2009. However, today the administration of a polysaccharide conjugate vaccine is recommended, followed by the 23-valent polysaccharidic vaccine at a later date [10].

When new or reactivations of infections occur, they need to be diagnosed at an early stage. Recent technological advances have been made in this context. Since 2014, a multiplex polymerase chain reaction (PCR) on bronchoalveolar lavage samples (BAL) has been available in our institution [11]. Furthermore, for invasive aspergillosis infections, the determination of N-galactomannan levels in serum and BAL since 2008 facilitates diagnosis and patient management, as it is proportional to the fungal load and is of prognostic value [12].

Despite new diagnostic and management methods, clinicians are challenged with epidemiological changes, including a significant increase in antibiotic-resistant Gram-negative bacilli (GNB) [13], and rare fungal infections [14], as well as viral epidemics such as H1N1 [15] and more recently the Severe Acute Respiratory Syndrome Coronavirus-2 (SARS-CoV-2). Only one recent study, on PT infections in a cohort of 2761 Swiss SOT patients [16], has accounted for some of these changes. Due to the lack of recent Belgian data, (971 organs from deceased donors were donated in 2022 [17]) accounting for recent epidemiological changes and the availability of novel vaccines and/or diagnostic tools, we performed a retrospective study on SOT infections occurring during the first-year PT at Hôpital Universitaire de Bruxelles-Erasme (HUB-Erasme).

Our objective was to describe infections up until one-year PT in Belgian SOT patients. The second objective was to identify risk factors for developing these infections so as to better prevent them in the future. 

## 2. Materials and Methods

### 2.1. Inclusion and General Principles

We conducted a retrospective descriptive study of adult SOT recipients at HUB-Erasme, a 1048-bed university hospital located in Brussels [18], Belgium. The observation period was between 1 January 2018 and 31 December 2019. Any patient over 18 years of age who underwent transplantation of one or more organs such as lungs, heart, kidney, liver and/or pancreas was included. Records were reviewed for a period of one year following transplantation. We excluded records of patients considered incomplete, defined as patients who were alive but followed up for less than 6 months. To identify patients for inclusion, a list of patients was obtained from the transplant coordination secretariat. All data were extracted from computerized medical records.

We collected data prior to transplantation, such as demographic characteristics (age and sex), comorbidities, pathologies leading to transplantation, pneumococcal vaccination status, history of hepatitis B and C infection, diagnosis and treatment of latent infections (tuberculosis, and strongyloides infection), and immunosuppressive drugs, and/or antibiotics received in the month prior to transplantation.

The Belgian Health Council (BHC) previously recommended (up until 2023) vaccinating patients against *S. pneumoniae* with Prevenar-13^®^ followed by Pneumovax-23^®^ [19], with at least an 8-week interval between vaccine administrations. In our study, despite the BHC recommendations, we considered a patient as vaccinated against *S. pneumoniae* if at least one of the two vaccines were administered.

The diagnosis of latent tuberculosis was retained if an interferon-gamma release assay test and/or a tuberculin skin test was positive. Imaging was not considered, due to a significant number of missing protocols. A strongyloides infection was diagnosed based on serology and/or positive stool tests.

We collected operation-specific data, such as immediate pre-SOT hospitalization, transplant emergency, donor after circulatory death (DCD), donor after brain death (DBD or living donor), operative time, volume of blood loss, organ ischemia time, and intraoperative antibiotic prophylaxis received.

We recorded multidrug-resistant (MDR) bacterial colonization and the Cytalomegavirus (CMV), the Eptsein–Barr virus (EBV) and the toxoplasmosis serological status of donors and recipients. We also recorded the history of active hepatitis C infection (positive HCV serology accompanied with or without a detectable viral load), and hepatitis B infection (presence of HBs antigen (HBsAg), or positive anti-HBc serology).

For PT data, we collected CMV and pneumocystis (PCP) prophylaxis, immunosuppressive therapy (all treatments received PT), influenza vaccination, and neutropenic status. MDR bacteria were defined as a bacteria resistant to at least one antibiotic in at least three different classes of antibiotics [20]. Among the MDR bacteria, we recorded data on methicillin-resistant *Staphylococcus aureus* (MRSA), extended-spectrum beta-lactamase producing enterobacteriaceae (ESBL), carbapenemase-producing enterobacteriaceae (CPE) and vancomycin-resistant *Enterococcus faecium* (VRE).

Regarding serological status (CMV, EBV, toxoplasmosis), a mismatch was defined as positive serology in the donor, yet negative in the recipient. The standard of care in Belgium is to give CMV prophylaxis with valganciclovir to all patients, unless both donor and recipient are CMV negative. Prophylaxis was considered adequate if the duration was at least 3 months [21]. For PCP, prophylaxis with Trimethoprim-sulfamethoxazole or pentacarinat aerosol was also considered adequate if its’ duration was greater than or equal to 3 months for the kidneys, 6 months for the livers and hearts and for the entire 12 months for the lungs [22,23,24].

We considered a patient neutropenic if the neutrophil count was less than 500/mm^3^. We recorded the number of episodes and duration of neutropenia below 500/mm^3^.

Concerning infections, we recorded the number of episodes per patient. For each episode, we recorded the timing (i.e., early, intermediate, or late), and the site of the infection, as well as the infecting pathogen(s) and its antibiogram if it was bacteria.

Finally, we collected data on the evolution of the patients over the year: the duration of the first hospitalization, the number of re-hospitalizations (defined as a stay of longer than 24 h), their duration, death and its cause.

We assessed an infected patient according to the clinician’s opinion, based on a combination of clinical, biological, microbiological documentation and/or imaging. If a microbiological sample (other than a blood culture) was positive for a bacteria, parasite, or fungus, yet no anti-infective treatment was administered, we concluded that the sample reflected colonization or contamination. A viral infection was retained if the patient had symptoms and the viral culture or PCR test was positive. 

Several pathogens could be responsible for the same infection. 

In terms of severity of infection, we classified infections into non-severe infections, or patients with sepsis or septic shock. Sepsis is defined as organ dysfunction caused by an inappropriate host response to infection. Organ dysfunction can be identified as an acute change in total Sequential Organ Failure Assessment (SOFA) score ≥ 2 consequent to the infection. Septic shock is defined as a sepsis requiring vasopressors to maintain the blood pressure ≥ 65 mmHg and a lactate > 2 mmol/L [25].

### 2.2. Statistical Analysis

Descriptive statistics were used to summarize our data. Discrete data were expressed as counts and percentages, and continuous variables by mean and standard deviation when the distribution followed a normal distribution, otherwise by a median with the minimum and maximum. An analysis of variance (ANOVA) was performed to compare the 4 groups of heart-, lung-, kidney- and liver-transplant patients.

Risk factors for presenting a first infectious episode and for dying were searched for. Univariate logistic regression analysis was performed with each potential explanatory variable. The variables considered for evaluation were as follows: the organ(s) transplanted, the number of transplanted organs, age over 40 years old, sex, hypertension, diabetes, obesity, former smoker, pre-transplant immunosuppressive treatment, CMV mismatch (R−/D+), CMV negative (R−/D−), bacterial colonization pre-transplantation, emergency transplant, volume of blood loss, pre-transplant hospitalization, no adequate CMV prophylaxis, no influenza vaccination, no pneumococcal vaccination and neutropenia. Furthermore, we also evaluated severity of infection as a risk factor for death.

Odds ratios were presented with their 95% confidence intervals. All predictors associated with an outcome of a *p* < 0.25 were considered for multivariate analyses. A mixed predictor selection procedure was then applied. The choice of the selected model was based on statistical significance. The fit of the model to the data was checked with the Hosmer–Lemeshow test. The absence of collinearity between predictors was checked with the variance inflation factor. The significance level was set at 0.05. Statistical analyses were performed with Stata/MP 14.1 and GraphPad Prism 9.1.0.

### 2.3. Ethics Committee

The study was approved by the Ethics Committee of HUB-Erasme in November 2020, under the reference P2021/004.

## 3. Results

### 3.1. Inclusion and Distribution of Patients

We included 231 transplant patients (Figure 1). The most frequently transplanted organs were kidneys and livers. Only two pancreases were transplanted and always simultaneously with a kidney.

### 3.2. Patient Demographics

Table 1 summarizes the demographic characteristics of our patients. Most subjects were male, with a median age of 57 years old, with no significant differences between transplanted organs. Comorbidities depended on the organ failure, but the most frequent were arterial hypertension and renal insufficiency.

The most frequent pathologies leading to transplantation (Table 2) were dilated heart disease for heart transplantation, hepatocarcinoma for liver transplantation, emphysema for lung transplantation, and both glomerulonephritis and nephroangiosclerosis for kidney transplantation.

Table 3 shows the vaccines and treatments administered to patients during the pre-transplantation period. One hundred and twelve patient (48%) were vaccinated against *S. pneumoniae* before transplantation. Latent tuberculosis was found in 21 patients (9%), all of whom received treatment. However, 113 patients (49%) had an unknown tuberculosis status.

Table 4 shows the pre-transplant serostatus (CMV, EBV and toxoplasmosis) and colonization of patients in our cohort. Forty patients (17%) had a mismatch.

Fifty-nine patients were colonized with bacteria (26%) before transplantation, 24 (10%) of which were MDR bacteria, identified primarily in the respiratory and rectal samples. Thirty-seven patients (16%) received an organ colonized with a bacteria, seven of whom received an organ colonized with a MDR bacteria. The majority of colonized organs were the lungs.

### 3.3. Characteristics Related to the Transplant Operation

Table 5 summarizes the surgical characteristics of our cohort. There were 20 emergency transplants; the majority were for liver transplantations. The most significant blood loss was recorded for liver transplantations and the longest operative times for lung transplantations.

### 3.4. Characteristics of the Post-SOT Patients

In Table 6, we present PT characteristics of our cohort. Concerning immunosuppressive treatments, no liver-transplant recipient received thymoglobulin, and no lung-transplant recipient received basiliximab, ciclosporine or everolimus.

We note that 192 out of the 197 patients who were supposed to receive valganciclovir prophylaxis against CMV, truly received it. Ten out of these 192 patients did not receive adequate prophylaxis due to side effects. Indeed, 27 patients (11%) presented an episode of neutropenia during the PT year with a median duration of 10 days.

### 3.5. Characteristics of the Infections

Table 7 provides data concerning the patients infected post-SOT. There were 143 patients (62%) who presented at least 1 infection over the year, with a total number of 305 infectious episodes, corresponding to a median of 2 episodes per patient (1–13).

Figure 2, Figure 3 and Figure 4 illustrate the first, second and third infectious episodes in function of time and according to each organ transplanted, respectively. During the first three episodes of infection, lung transplant patients were the most frequently infected, differing significantly from liver transplant patients who presented the fewest infectious episodes. Almost 25% of the patients had 2 infections during the first 6 months PT and less than 15% of the patients had 3 infectious episodes during the year.

The majority of infectious episodes, all organs combined, occurred during the intermediate period.

The site of infection varied according to the organ transplanted, as illustrated in Figure 5. The majority of infections were of respiratory or digestive origin.

Figure 6 and Figure 7 show the distribution of infecting pathogens according to the transplanted organ. In 44/305 (14%) cases, no pathogens were identified.

During each period, and throughout the entire year, infections were most often due to bacteria (n = 235, 62%) and particularly to GNB (n = 140, 60%), followed by viruses (CMV being the most prevalent) and then fungi. Among the GNB, *Escherichia coli*, *Klebsiella pneumoniae* and *Pseudomonas aeruginosa* were the most common, and of the Gram-positive cocci, *Enterococcus faecium* and *Staphylococcus aureus* were the most frequent. Three patients developed a *S. pneumoniae* pneumonia (without associated bacteremia), two of whom were vaccinated against *S. pneumoniae*. A total of 18% (N = 41/231) of patients were infected with MDR bacteria.

### 3.6. Evolution over the Year

Table 8 describes the evolution during the year after SOT. Nineteen patients (8%) died over the year following their SOT; 11 deaths (5%) were attributed to an infectious cause. The main infections responsible for these deaths were bacteremia (5/11) of pulmonary origin (3/5), one of urinary origin and one of digestive origin.

### 3.7. Risk Factors for Infection

We tried to identify risk factors for developing at least one infection during the PT year (Table 9). The following factors were considered: transplanted organs, number of transplanted organs, age over 40 years old, sex, hypertension, diabetes, obesity, former smoker, pre-transplant immunosuppressive treatment, CMV mismatch (R−/D+), CMV negative (R−/D−), bacterial colonization pre-transplantation, emergency transplant, volume of blood loss, pre-transplant hospitalization, no adequate CMV prophylaxis, no influenza vaccination, no pneumococcal vaccination and neutropenia.

The univariate, followed by multivariate analysis (Table 10) revealed five independent variables associated with the risk of infection. Three variables positively influenced the risk of infection: lung transplantation, CMV mismatch and neutropenia. On the other hand, liver transplantation and obesity protected against infection.

### 3.8. Risk Factor for Death

Finally, we explored risk factors for dying. The same potential risk factors for infection were explored, as well as the severity of the infection. The risk factors for death found by multivariate analysis (Table 11), after univariate analysis, showed that non-vaccination against *S. pneumoniae*, inadequate CMV prophylaxis and severity of infection were the main risk factors for death.

## 4. Discussion

We describe a cohort of 231 patients who underwent an SOT in a Belgian University Hospital between 2018 and 2019 and who were followed up for one year. This cohort was mainly composed of men in their fifties, with at least one comorbidity. In our center, the kidney was the most frequently transplanted organ. Nearly two thirds of patients developed at least one infection and more than 50% presented at least two infectious episodes during the PT period. The respiratory tract was the primary site of infection, accounting for 30% of all infections. The most frequent pathogens were bacteria, accounting for 60% of infections, of which 35% were caused by an MDR bacteria. We identified several risk factors for developing the first infection: lung transplantation, CMV mismatch and neutropenia. Protective factors were liver transplantation and obesity. Almost one in ten patients died during the year PT, of which deaths more than 50% were due to an infectious etiology. Risk factors for death were severity of the infection, inadequate CMV prophylaxis and non-vaccination against *S. pneumoniae*.

### 4.1. Incidence and Epidemiology of Infections

Our observations are consistent with the only recent European study on a Swiss cohort in terms of proportion of transplanted organs, demographic characteristics, incidence of infections (55% in the Swiss cohort) and bacterial predominance of infections. As in our cohort, most infections occurred during the first six months post-SOT [16].

However, our observations concerning viral infections are discordant. In our study, CMV was the main virus responsible for infections. For the Swiss, CMV was rarely an identified pathogen, despite the fact that only 50% of the cohort received a valganciclovir/ganciclovir prophylaxis. The differences could be due to variations in local epidemiology.

### 4.2. Risk Factors for PT Infections

The risk of developing an infection depends strongly on the organ transplanted and the underlying pathology [1]. In our study, lung transplant recipients had the highest risk and liver recipients the lowest risk of developing a PT infection. Indeed, 94% of our lung transplant patients developed an infection during the first year PT, while only 40% of our liver transplant patients did. These results concord with a review article on infections in lung transplant recipients [26]. Our observations could partially be explained by differences in immunosuppressive treatment in function of the organ transplanted. Indeed, none of the liver transplant recipients received thymoglobulin, whereas all lung transplant recipients did.

The other two risk factors identified for infection were CMV mismatch, and neutropenia. It is well known that CMV infection is responsible for complications such as graft loss, other opportunistic infections, and death [2,21]. In our study, CMV was indirectly responsible for the risk of infection via CMV-mismatch status, as we know that mismatch is a risk factor for developing CMV infections [2,8,21,27]. Indeed, 75% of patients with a CMV mismatch developed a CMV infection in our study. Furthermore, the recommended prophylaxis for CMV is Ganciclovir or Valganciclovir [28]. However, these drugs have hematological side effects, such as neutropenia [29]. In our study, 10 patients had to stop their CMV prophylaxis due to neutropenia.

In addition to liver transplantation, obesity at the time of transplantation was identified as a protective factor against PT infections. Obesity is associated with an increased risk of hospital-acquired infections in the general population [30]. However, there are controversial findings regarding morbidity and mortality in obese SOT patients [31,32]. In our study, we did not collect data on exact body-mass indexes of patients, but patients with a BMI > 35 kg/m^2^ are not accepted for SOT in our center. Previous studies have shown that PT complications are not increased in patients with a BMI between 30 and 34.9 kg/m^2^ compared to non-obese patients, contrary to patients with a BMI > 35 kg/m^2^ [33]. Therefore, our results could be explained by confounding factors, as obesity at the time of the transplantation may reflect a better general health and nutritional status compared to other non-obese patients with end-stage organ failure.

### 4.3. Risk Factors for Death

We observed a 5% mortality due to infectious causes in our cohort, which is higher than the 2% reported in the Swiss cohort [16]. Severity of infection was an important risk factor for mortality identified in our study. This is not surprising, as a mortality rate of 25–30% for sepsis and 40–60% for septic shock is observed in the general population [34]. Furthermore, the risk of developing sepsis depends on the patient’s risk factors, and immunosuppression is one of them [34].

Another identified risk factor for death was inadequate CMV prophylaxis. As explained earlier, the inadequate CMV prophylaxis was due to hematological toxicity resulting in neutropenia with no other therapeutic options for prophylaxis.

The last, but not least, important risk factor for death identified in our study was to not be vaccinated against *S. pneumoniae*, as was observed in 52% of our patients. A recent point-prevalence study on pneumonias in SOT patients in Spain and Italy reported a community-acquired pneumonia (CAP) incidence of 40.7% [35]. *S. pneumoniae* remains the leading pathogen for CAP worldwide and vaccination against this pathogen has been shown to decrease the risk of developing CAP [36]. Therefore, although *S. pneumoniae* was not a risk factor for infections, nor a leading infecting pathogen, nor a leading cause of death in our cohort, improving vaccination against this pathogen appears to be essential.

### 4.4. Strengths and Limitations

The recent and important novel data on infectious episodes up until one year PT from a transplant center in Belgium are strengths of our study. Nevertheless, this study has certain limitations. Firstly, it is a retrospective, monocentric study. Secondly, transplant patients are not followed exclusively in our center during the PT period. Therefore, the medical charts may not be complete concerning all infectious episodes (i.e., simple infections not needing hospitalizations). Thirdly, we were unable to identify and encode episodes of graft rejection, nor the dynamics of immunosuppressive treatments over time; these factors may have influenced the occurrence of infections. Fourthly, we did not assess whether intraoperative antibiotic prophylaxis received for transplantation was adequate, despite that 24 of our patients were colonized by MDR bacteria, and 7 received an organ colonized by an MDR bacteria. Finally, although hepatitis E infections are re-emerging [37], no data were collected on serological status for hepatitis E in our cohort.

## 5. Conclusions

SOT is a well-accepted treatment for end-stage organ failure. In our retrospective, monocentric cohort study, despite advances in patient management, infectious episodes were very frequent, and the cause of over 50% of deaths during the first year PT. Nevertheless, mortality could be reduced by addressing modifiable risk factors. Given the risk of death associated with non-vaccination against *S. pneumoniae*, we advocate better pre-transplant prevention with systematic vaccination against this pathogen. Considering the risk of death associated with inadequate anti-CMV prophylaxis, particularly due to hematological toxicity of ganciclovir or valganciclovir, we advocate making drugs such as Maribavir more widely accessible to SOT patients. Indeed, this drug has already proven to be as effective in pre-emptive treatment of CMV reactivations in SOT patients while causing less neutropenia than valganciclovir [38].

Finally, a significant number of patients in our cohort were colonized and/or infected with an MDR bacteria. Although infections due to MDR bacteria were not identified as a risk factor for death in our study, these infections are a growing problem worldwide, and continuous efforts to fight antimicrobial resistance and/or develop new antibiotics or therapeutics active against these pathogens are essential [39].

## Figures and Tables

**Figure 1 microorganisms-12-00755-f001:**
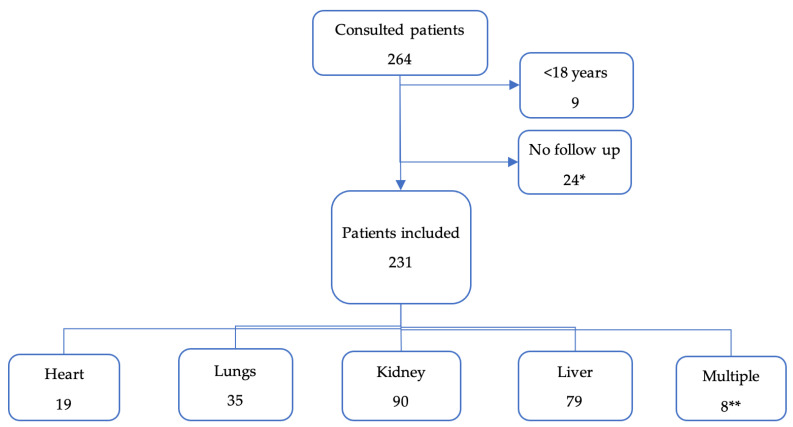
Flow chart of patient inclusions, and organs transplanted. * All of them were kidney transplant, except for one patient (liver). ** Multiple: 3 livers + kidneys; 1 liver + lungs; 2 pancreas + kidneys; 1 heart + lungs; 1 heart + kidney.

**Figure 2 microorganisms-12-00755-f002:**
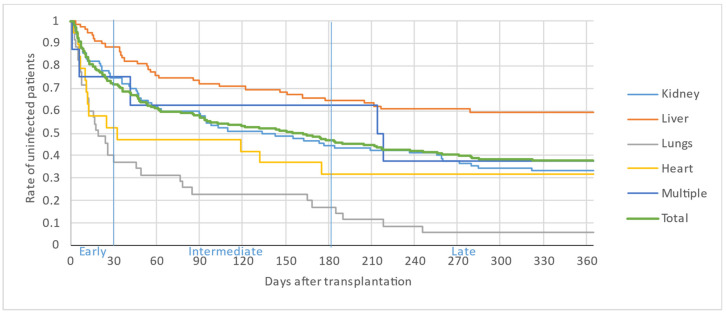
First infectious episode by organ.

**Figure 3 microorganisms-12-00755-f003:**
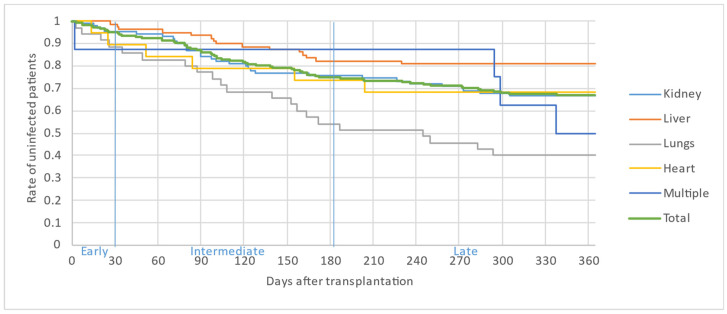
Second infectious episode by organ.

**Figure 4 microorganisms-12-00755-f004:**
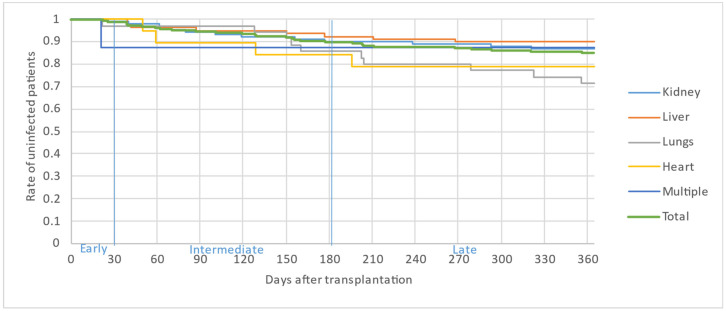
Third infectious episode by organ.

**Figure 5 microorganisms-12-00755-f005:**
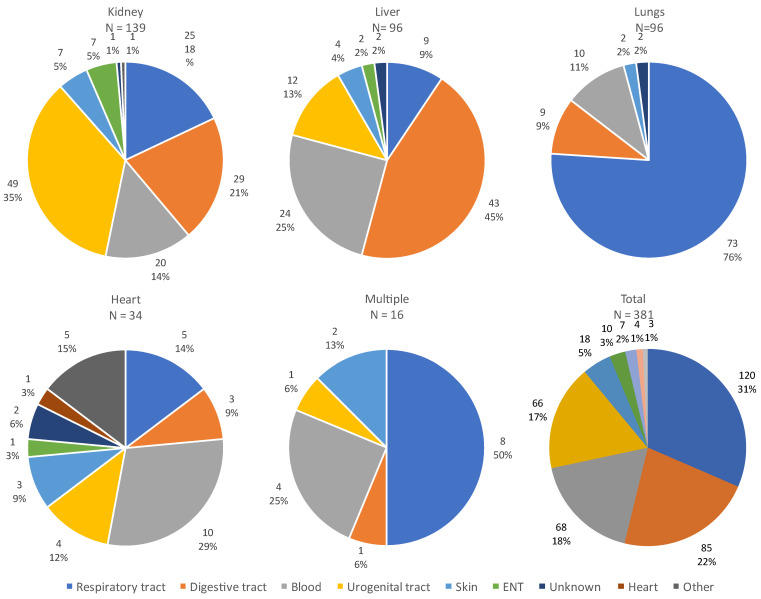
Sites of all infections according to the transplanted organ. **Abbreviations**: **ENT:** ear, nose and throat.

**Figure 6 microorganisms-12-00755-f006:**
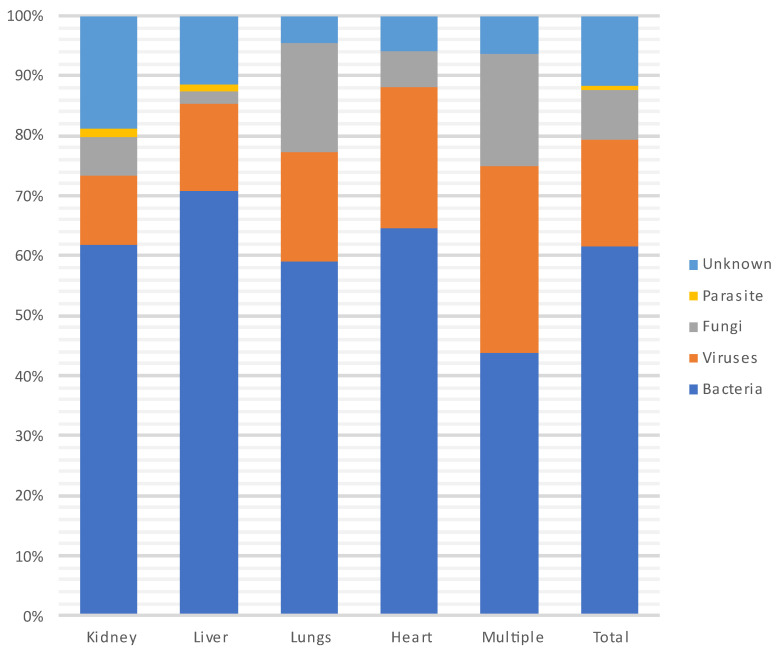
Distribution of infecting pathogens according to the transplanted organ.

**Figure 7 microorganisms-12-00755-f007:**
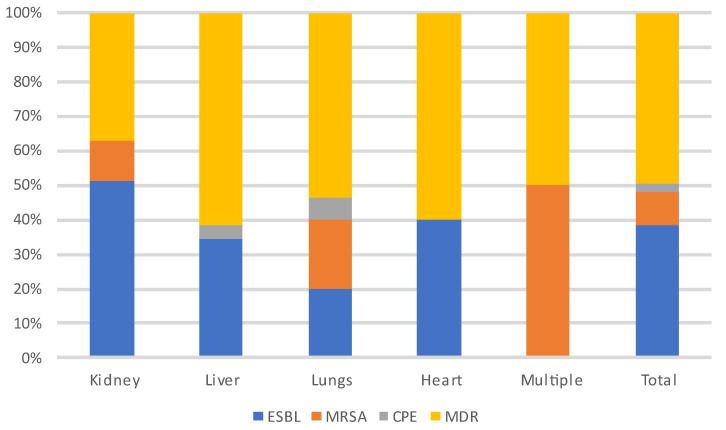
Distribution of MDR-infecting pathogens according to the transplanted organ. **Abbreviations: ESBL**: extended-spectrum beta-lactamase producing enterobacteriaceae; **MRSA**: methicillin-resistant *Staphylococcus aureus*; **CPE**: carbapenemase-producing enterobacteriaceae; **MDR**: multi-drug-resistant bacteria.

**Table 1 microorganisms-12-00755-t001:** Demographic characteristics and comorbidities in function of the transplanted organ.

	Heart n = 19	Lungs n = 35	Kidney n = 90	Liver n = 79	Multiple n = 8	Total n = 231	*p*-Value
Age	56 (29–65)	57 (18–65)	56.5 (21–74)	58 (28–72)	41 (24–59)	57 (18–74)	0.1716
Sex (male)	16 (84)	21 (60)	58 (64)	57 (72)	5 (63)	157 (68)	0.2162
History of smoker	8 (42)	16 (46)	25 (28)	21 (27)	2 (25)	72 (31)	0.1258
Arterial hypertension	8 (42)	9 (26)	76 (84)	30 (38)	3 (38)	126 (54)	**<0.0001**
Obesity	3 (16)	2 (6)	16 (18)	16 (20)	3 (38)	40 (17)	0.0819
Diabetes	5 (26)	8 (23)	32 (36)	22 (28)	3 (38)	70 (30)	0.3498
Renal insufficiency	6 (32)	1 (3)	90 (100)	13 (16)	6 (75)	116 (50)	**<0.0001**
COPD	0 (0)	14 (40)	4 (4)	6 (8)	1 (13)	25 (11)	**<0.0001**
Cardiopathy	19 (100)	8 (23)	24 (27)	7 (9)	3 (38)	61 (27)	**<0.0001**
LVAD	13 (68)	0 (0)	0 (0)	0 (0)	0 (0)	13 (6)	**<0.0001**

Qualitative data are presented as number of cases (% within group), quantitative data as median (min–max). The chosen threshold of statistical significance is 0.05. Significant *p*-values are in bold. **Abbreviations: COPD**: chronic obstructive pulmonary disease; **LVAD**: left-ventricular assist device.

**Table 2 microorganisms-12-00755-t002:** Pathology leading to transplantation.

**Cardiopathy: Heart transplantation**	n = 21
Dilated heart disease	10 (48)
Ischemic heart disease	6 (29)
Other	5 (24)
**Respiratory failure: Lung transplantation**	n = 37
Emphysema	13 (35)
Cystic fibrosis	9 (24)
Fibrosis	8 (22)
Pulmonary arterial hypertension	4 (11)
Other	3 (8)
**Kidney failure: Kidney transplantation**	n = 96
Glomerulonephritis	24 (25)
Hypertensive/diabetic nephroangiosclerosis	24 (25)
Chronic tubulointerstial nephropathy	5 (5)
Polycystic kidney disease	5 (5)
Other	38 (21)
**Liver failure: Liver transplantation**	n = 83
*Chronic*	77 (93)
Hepatocarcinoma	38 (46)
Cirrhosis	34 (41)
Other	5 (6)
*Acute*	6 (7)
Drug-induced *	2 (2)
Flare of hepatitis B	2 (2)
Other	(2)

* No acetaminophen-induced.

**Table 3 microorganisms-12-00755-t003:** Vaccines, latent infections and treatments administered pre-transplantation, according to the transplanted organ.

	Heart n = 19	Lungs n = 35	Kidney n = 90	Liver n = 79	Multiple n = 8	Total n = 231	*p*-Value
Pneumococcal vaccination	1 (5)	19 (54)	66 (73)	22 (28)	4 (50)	112 (48)	0.8932
Latent tuberculosis							0.2665
Yes	1 (5)	0 (0)	12 (13)	6 (8)	2 (25)	21 (9)	
No	5 (26)	7 (20)	34 (38)	47 (59)	4 (50)	97 (42)	
Unknown	13 (68)	28 (80)	44 (49)	26 (33)	2 (25)	113 (49)	
Strongyloides infection							-
Yes	0 (0)	0 (0)	0 (0)	0 (0)	0 (0)	0 (0)	
Unknown	14 (74)	35 (100)	75 (83)	78 (99)	7 (88)	209 (90)	
Hepatitis B	0 (0)	1 (3)	5 (6)	8 (10)	0 (0)	14 (6)	
Hepatitis C	1 (5)	0 (0)	0 (0)	11 (14)	0 (0)	12 (5)	
Antibacterial treatment	1 (5)	18 (51)	5 (6)	13 (16)	1 (13)	38 (16)	**<0.0001**
Azithromycin	0 (0)	14 (40)	2 (2)	0 (0)	0 (0)	16 (7)	
Cotrimoxazole	1 (5)	5 (14)	3 (3)	2 (3)	1 (13)	12 (5)	
Other	0 (0)	4 (11)	1 (1)	11 (14)	0 (0)	16 (7)	
Immunosuppressive treatment							**0.0053**
Yes	0 (0)	10 (29)	22 (24)	10 (13)	1 (13)	43 (19)	
Unknown	0 (0)	0 (0)	2 (2)	1 (1)	1 (13)	4 (2)	

Data are presented as number of cases (% intra-group). The chosen threshold of statistical significance is 0.05. Significant *p*-values are in bold.

**Table 4 microorganisms-12-00755-t004:** Pre-transplant serostatus and colonization according to the transplanted organ.

	Heart n = 19	Lungs n = 35	Kidney n = 90	Liver n = 79	Multiple n = 8	Total n = 231	*p*-Value
Positive recipient serology							
CMV	18 (95)	15 (43)	70 (78)	48 (61)	5 (63)	156 (68)	**<0.0001**
Toxoplasmosis	11 (58)	16 (46)	58 (64)	54 (68)	6 (75)	145 (63)	0.199
EBV	18 (95)	35 (100)	89 (99)	74 (94)	8 (100)	224 (97)	0.3877
Mismatch CMV (R−/D+)	1 (5)	6 (17)	13 (14)	18 (23)	2 (25)	40 (17)	0.1152
Negative CMV serology (R−/D−)	0 (0)	14 (40)	6 (7)	13 (16)	1 (13)	34 (15)	**<0.0001**
Bacterial colonization of recipient	9 (47)	18 (51)	5 (6)	24 (30)	3 (38)	59 (26)	**<0.0001**
Colonization of MDR bacteria	1 (5)	10 (29)	4 (4)	7 (9)	2 (25)	24 (10)	
GNB	1 (5)	9 (26)	3 (3)	5 (6)	2 (25)	20 (9)	
ESBL	0 (0)	1 (3)	1 (1)	1 (1)	0 (0)	3 (1)	
CPE	0 (0)	0 (0)	0 (0)	1 (1)	1 (13)	2 (1)	
GPC	0 (0)	3 (9)	1 (1)	3 (4)	1 (13)	8 (3)	
*E. faecium* Ampi-R	0 (0)	0 (0)	0 (0)	3 (4)	0 (0)	3 (1)	
VRE	0 (0)	0 (0)	0 (0)	0 (0)	0 (0)	0 (0)	
MRSA	0 (0)	3 (9)	1 (1)	0 (0)	1 (13)	5 (2)	
Unknown	3 (16)	7 (20)	1 (1)	6 (8)	2 (25)	19 (8)	
Location of MDR bacterial colonization							
Lungs	0 (0)	8 (23)	0 (0)	1 (1)	1 (13)	10 (4)	
Rectal swabs	1 (5)	0 (0)	2 (2)	1 (1)	2 (25)	6 (3)	
Urine	0 (0)	0 (0)	0 (0)	2 (3)	0 (0)	2 (1)	
Donor bacterial colonization	4 (21)	22 (63)	4 (4)	1 (1)	6 (75)	37 (16)	**<0.0001**
MDR bacteria colonization	2 (11)	4 (11)	0 (0)	1 (1)	0 (0)	7 (3)	

Data are presented as number of cases (% intra-group). The statistical significance level chosen is 0.05. Significant *p*-values are in bold. **Abbreviations: CMV**: Cytomegalovirus; **EBV**: Epstein–Barr virus; **R**: recipient; **D**: donor; **MDR**: multidrug-resistant; **GNB**: Gram-negative bacillus; **ESBL**: extended-spectrum beta-lactamase-producing enterobacteriaceae; **CPE**: carbapenemase-producing enterobacteriaceae; **GPC**: Gram-positive coccus; **Ampi-R**: Ampicillin-resistant; **VRE**: vancomycin-resistant Enterococci; **MRSA**: methicilin-resistant *Staphylococcus aureus*.

**Table 5 microorganisms-12-00755-t005:** Information about the transplant surgery according to the transplanted organ.

	Heart n = 19	Lungs n = 35	Kidney n = 90	Liver n = 79	Multiple n = 8	Total n = 231	*p*-Value
**Donor status**							
Cadaveric: DCD	1 (5)	6 (17)	22 (24)	16 (20)	1 (13)	46 (20)	
Cadaveric: DBD	18 (95)	20 (57)	58 (64)	55 (70)	6 (75)	157 (68)	**0.0016**
Living	0 (0)	0 (0)	10 (11)	0 (0)	0 (0)	10 (4)	
Unknown	0 (0)	9 (26)	0 (0)	6 (8)	1 (13)	16 (7)	
Emergency transplant	5 (26)	2 (6)	0 (0)	12 (15)	1 (13)	20 (9)	**0.0037**
Blood loss (L)	1.05 (0.2–6)	0.9 (0.25–15)	0.3 (0.05–2.5)	1.75 (0.1–10.5)	1.3 (0.9–3.2)	1 (0.05–15)	**<0.0001**
Operating time (h)	7.7 (4.9–15.9)	8.0 (6.0–19.3)	2.5 (0.8–4.0)	5.5 (4.1–9.8)	7.1 (4.9–22.2)	5.1 (0.8–22.2)	**<0.0001**
Time of ischemia (h)	2.7 (1.8–4.7)	7.0 (1.8–10.8)	12.8 (1.9–25.9)	6.4 (2.2–12.0)	10.9 (10.4–11.4)	7.6 (1.8–25.9)	**<0.0001**
Length of hospitalization (d)	53 (22–157)	27 (2–139)	10 (5–37)	15 (1–182)	21.5 (9–92)	15 (1–182)	**<0.0001**

Qualitative data are presented as number of cases (% within group), quantitative data as median (min–max). The statistical significance level chosen is 0.05. Significant *p*-values are in bold. **Abbreviations**: **DCD**: donor after circulatory death; **DBD:** donor after brain death; **L**: Liter; **h**: hour; **d**: day.

**Table 6 microorganisms-12-00755-t006:** Post-transplant characteristics: prophylaxis, vaccination, immunosuppressive therapy and neutropenia according to the transplanted organ.

	Heart n = 19	Lungs n = 35	Kidney n = 90	Liver n = 79	Multiple n = 8	Total n = 231	*p*-Value
**Intra-operative antibacterial prophylaxis**							
Yes	19 (100)	34 (97)	65 (72)	73 (92)	8 (100)	199 (86)	
Unknown	0 (0)	1 (3)	25 (28)	6 (8)	0 (0)	32 (14)	
**CMV prophylaxis**							**0.0019**
Yes	17 (89)	20 (57)	83 (92)	65 (82)	7 (88)	192 (83)	
Unknown	1 (5)	1 (3)	0 (0)	0 (0)	0 (0)	2 (1)	
Adequate prophylaxis	16 (84)	17 (49)	78 (87)	59 (75)	6 (75)	176 (76)	
**PCP prophylaxis**							0.0628
Yes	17 (89)	34 (97)	90 (100)	75 (95)	8 (100)	224 (97)	
Unknown	1 (5)	0 (0)	0 (0)	0 (0)	0 (0)	1 (0)	
Adequate prophylaxis	12 (63)	32 (91)	89 (99)	52 (66)	5 (63)	190 (82)	
**Influenza vaccination PT**							**0.0002**
Yes	1 (5)	14 (40)	33 (37)	23 (29)	3 (38)	74 (32)	
Unknown	18 (95)	21 (60)	57 (63)	56 (71)	5 (63)	157 (68)	
**All immunosuppressive treatment received post-SOT**							
Thymoglobulin	18 (95)	35 (100)	46 (51)	0 (0)	5 (63)	104 (45)	
Basiliximab	1 (5)	0 (0)	45 (50)	20 (25)	3 (38)	69 (30)	
Methyl-prednisolone	19 (100)	35 (100)	90 (100)	77 (97)	8 (100)	229 (99)	
Mycophenolate mofetil	19 (100)	35 (100)	90 (100)	79 (100)	8 (100)	231 (100)	
Tacrolimus	16 (84)	34 (97)	90 (100)	78 (99)	7 (88)	225 (97)	
Ciclosporine	5 (26)	0 (0)	1 (1)	1 (1)	1 (13)	8 (3)	
Azathioprine	0 (0)	14 (40)	18 (20)	1 (1)	1 (13)	34 (15)	
Everolimus	1 (5)	0 (0)	8 (9)	23 (29)	0 (0)	32 (14)	
**Neutropenia**							
PNN < 500/mm^3^	2 (50)	2 (29)	10 (63)	13 (48)	0 (0)	27 (50)	0.34
Number of events	1 (1–1)	1 (1–1)	1 (1–2)	1 (1–2)	0 (0–0)	1 (1–2)	
Duration (d)	7 (8–10)	13 (7–15)	13 (2–18)	11 (2–315)	0 (0–0)	10 (1–315)	

Data are presented as number of cases (% intra-group). The statistical significance level chosen is 0.05. Significant *p*-values are in bold. **Abbreviations**: **CMV**: Cytomegalovirus; **PCP**: Pneumocystis Jirovecii; **PT**: post-transplantation; **SOT**: solid-organ transplant; **PNN**: polynuclear neutrophils; **d**: days.

**Table 7 microorganisms-12-00755-t007:** Patients infected during the year post-transplantation according to the transplanted organ.

	Heart n = 19	Lungs n = 35	Kidney n = 90	Liver n = 79	Multiple n = 8	Total n = 231	*p*-Value
**Infections within the first month PT**							
Infected patients	9 (47)	22 (63)	23 (26)	9 (11)	2 (25)	65 (28)	**<0.0001**
Bacteria	8 (42)	20 (57)	19 (21)	7 (9)	2 (25)	56 (24)	
MDR bacteria	2 (11)	7 (20)	9 (10)	3 (4)	1 (13)	22 (10)	
Fungi	0 (0)	3 (9)	2 (2)	0 (0)	1 (13)	6 (3)	
Parasites	0 (0)	0 (0)	1 (1)	0 (0)	0 (0)	1 (0)	
Virus	0 (0)	1 (3)	3 (3)	0 (0)	0 (0)	4 (2)	
Unknown	1 (5)	1 (3)	2 (2)	2 (3)	0 (0)	6 (3)	
Severity	3 (16)	0 (0)	0 (0)	2 (3)	1 (13)	6 (3)	0.0598
Death	2 (11)	1 (3)	0 (0)	1 (1)	0 (0)	4 (2)	0.2438
**Infections between 1 and 6 months PT**							
Infected patients	8 (42)	17 (49)	39 (43)	24 (30)	2 (25)	90 (39)	0.2089
Bacteria	7 (37)	4 (11)	20 (22)	18 (23)	1 (13)	50 (22)	
MDR bacteria	2 (11)	3 (9)	8 (9)	10 (13)	0 (0)	23 (10)	
Fungi	2 (11)	8 (23)	5 (6)	2 (3)	1 (13)	18 (8)	
Parasite	0 (0)	0 (0)	1 (1)	1 (1)	0 (0)	2 (1)	
Virus	2 (11)	11 (31)	8 (9)	9 (11)	1 (13)	31 (13)	
Unknown	0 (0)	0 (0)	11 (12)	6 (8)	0 (0)	17 (7)	
Severity	0 (0)	1 (3)	3 (3)	6 (8)	0 (0)	10 (4)	0.37
Death	0 (0)	1 (3)	1 (1)	5 (6)	0 (0)	7 (3)	0.2158
**Infections after 6 months PT**							
Infected patients	4 (21)	12 (34)	25 (28)	7 (9)	3 (38)	51 (22)	**0.0083**
Bacteria	2 (11)	4 (11)	11 (12)	4 (5)	2 (25)	23 (10)	
MDR bacteria	1 (5)	3 (9)	6 (7)	3 (4)	1 (13)	14 (6)	
Fungi	0 (0)	5 (14)	1 (1)	0 (0)	0 (0)	6 (3)	
Parasite	0 (0)	0 (0)	0 (0)	0 (0)	0 (0)	0 (0)	
Virus	4 (21)	8 (23)	5 (6)	2 (3)	2 (25)	21 (9)	
Unknown	1 (5)	3 (9)	11 (12)	3 (4)	1 (13)	19 (8)	0.7422
Severity	0 (0)	0 (0)	2 (2)	1 (1)	0 (0)	3 (1)	0.6128
Death	0 (0)	0 (0)	0 (0)	0 (0)	0 (0)	0 (0)	
**Infections within one year PT**							
Infected patients	13 (68)	33 (94)	60 (67)	32 (41)	5 (63)	143 (62)	**<0.0001**
Number of episodes	1 (1–7)	2 (1–13)	1.5 (1–7)	1 (1–9)	2 (1–5)	2 (1–13)	0.8228
Bacteria	12 (63)	24 (69)	38 (42)	22 (28)	4 (50)	100 (43)	
MDR bacteria	2 (11)	10 (29)	16 (18)	11 (14)	2 (25)	41 (18)	
Fungi	2 (11)	16 (46)	8 (9)	2 (3)	1 (13)	29 (13)	
Parasite	0 (0)	0 (0)	2 (2)	1 (1)	0 (0)	3 (1)	
Virus	4 (21)	18 (51)	16 (18)	10 (13)	3 (38)	51 (22)	
Unknown	2 (11)	4 (11)	22 (24)	10 (13)	1 (13)	39 (17)	
Severity	3 (16)	1 (3)	5 (6)	8 (10)	1 (13)	18 (8)	
Death	2 (11)	2 (6)	1 (1)	6 (8)	0 (0)	11 (5)	

Qualitative data presented as number of cases (% within group), quantitative data presented as median (min–max). Statistical significance level at 0.05. Significant *p*-values are in bold. **Abbreviations**: **PT**: post-transplantation; **MDR**: multidrug-resistant.

**Table 8 microorganisms-12-00755-t008:** Evolution over the year post-transplantation and severity, according to the transplanted organ.

	Heart n = 19	Lungs n = 35	Kidney n = 90	Liver n = 79	Multiple n = 8	Total n = 231	*p*-Value
Patients with re-hospitalization	11 (58)	26 (74)	46 (51)	39 (49)	5 (63)	127 (55)	0.0762
Number of re-hospitalizations	2 (1–8)	1 (1–6)	1 (1–5)	1 (1–6)	3 (1–3)	1 (1–8)	0.2289
Length of re-hospitalization	3.5 (2–43)	6 (2–33)	7 (2–112)	6 (1–142)	4 (2–20)	6 (1–142)	0.3501
Deaths	3 (16)	4 (11)	2 (2)	9 (11)	1 (13)	19 (8)	
Infectious deaths	2 (11)	2 (6)	1 (1)	6 (8)	0 (0)	11 (5)	
Non-infectious deaths	1 (5)	1 (3)	0 (0)	2 (3)	1 (13)	5 (2)	
Unknown death	0 (0)	1 (3)	1 (1)	1 (1)	0 (0)	3 (1)	

Qualitative data are presented as number of cases (% within group), quantitative data as median (min–max). The statistical significance level chosen is 0.05.

**Table 9 microorganisms-12-00755-t009:** Risk factors for infection: results of univariate logistic regression.

Risk Factor	n	Number	OR [CI 95%]	*p*-Value
Transplanted organ	231			<0.001
**Heart**		19	1.00	
**Liver**		79	0.31 [0.11–0.91]	
**Multiple**		8	0.77 [0.14–4.33]	
**Lungs**		35	7.62 [1.36–45.71]	
**Kidney**		90	0.92 [0.32–2.67]	
Hypertension	231	126	1.56 [0.91–2.66]	0.10
Obesity	231	40	0.38 [0.19–0.76]	0.007
Former smoker	231	72	1.61 [0.89–2.92]	0.11
Pre-transplant immunosuppressive treatment	227	43	1.55 [0.76–3.17]	0.23
CMV mismatch	231	40	2.43 [1.10–5.39]	0.03
Emergency transplant	231	20	1.95 [0.68–5.55]	0.21
Pre-transplant hospitalisation	231	35	1.95 [0.87–4.38]	0.11
No adequate CMV prophylaxis	186	10	6.38 [0.79–51.45]	0.08
No influenza vaccination	207	133	0.62 [0.34–1.13]	0.12
No pneumococcal vaccination	231	119	1.02 [0.60–1.74]	0.93
Neutropenia	231	46	1.73 [0.85–3.51]	0.13

The statistical significance level chosen was 0.25. **Abbreviations**: **OR**: odds ratio; **CI**: confidence interval; **CMV:** cytomegalovirus.

**Table 10 microorganisms-12-00755-t010:** Risk factors for infection: multivariate logistic regression.

Risk Factor	n	Number	OR [CI 95%]	*p*-Value
Liver	231	79	0.21 [0.07–0.66]	0.007
Lung	231	35	6.80 [1.17–39.36]	0.032
Obesity	231	40	0.41 [0.19–0.89]	0.025
CMV mismatch	231	40	3.53 [1.45–8.64]	0.006
Neutropenia	231	46	2.87 [1.27–6.47]	0.011

The statistical significance level chosen was 0.05. **Abbreviations**: **OR**: odds ratio; **CI**: confidence interval.

**Table 11 microorganisms-12-00755-t011:** Risk factors for death: multivariate logistic regression.

Variable	n	Number	OR [CI 95%]	*p*-Value
Not vaccinated against *S. pneumoniae*	231	119	19.78 [2.59–150.87]	0.004
No adequate CMV prophylaxis	186	10	22.8 [4.86–107.04]	<0.001
Severity (sepsis or septic shock)	231	18	28.33 [9.02–89.00]	<0.001

The statistical significance level chosen is 0.05. **Abbreviations**: **OR**: odds ratio; **CI**: confidence interval; **CMV**: cytomegalovirus.

## Data Availability

Data are contained within the article.

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
