# Peer review of "Monocentric, Retrospective Study on Infectious Complications within One Year after Solid-Organ Transplantation at a Belgian University Hospital"

_microorganisms, 2024, doi:10.3390/microorganisms12040755_

Round 1

Reviewer 1 Report

Comments and Suggestions for Authors

Authors present an interseting analysis of post-transplant infectious complications. Generally, article is well prepared and worth publishing. I would suggest some changes:

1. A low vaccination rate for S. pneumoniae and influenza. How frequent were S. pneumoniae and influenza infections in the cohort? Could you prepare an analysis of vaccinated vs unvaccinated patients, at least in regard to S. pneumoniae?

2. Lines 86-87: is that the period of transplantation or the period of observation? Please clearly indicate.

There are several typos in the text, please double-check and correct.

Author Response

Review 1 :

1. A low vaccination rate for S. pneumoniae and influenza. How frequent were S. pneumoniae and influenza infections in the cohort? Could you prepare an analysis of vaccinated vs unvaccinated patients, at least in regard to S. pneumonia?

Thank you for your comment.
We had 3 patients who developed a pneumonia due to Streptococcus pneumoniae  (without associated bacteremia), 2 of whom were vaccinated against S. pneumoniae.
Concerning influenza infections, there were 7 identified individuals with Influenza infections.  This information has been added into the manuscript. 

2. Lines 86-87: is that the period of transplantation or the period of observation? Please clearly indicate

Thank you for your comment.  I apologize for the lack of clarity. We collected transplant data during 2018 and 2019 and each patient was followed for 1 year. We have modified our text to try to add clarity.

3. There are several typos in the text, please double-check and correct

Thank you once again for your comment. We have verified the text and made adaptations.

Reviewer 2 Report

Comments and Suggestions for Authors

- In European countries, chronic infection of hepatitis E virus in SOT patients under immunosuppressive treatment has been a concern. Therefore, mentioning or inclusion of such cases in this study would be necessary.

- In view of this data (Table 3), chronic Viral hepatitis caused by HBV, HCV and HEV need to be mentioned in Introduction as well as Methodology sections.

- the English language should be reviewed.

For detailed comments (highlighted), please see the attached manuscript file (.pdf).

Comments on the Quality of English Language

Needs to be improved.

Author Response

Review 2 :

  1.  In European countries, chronic infection of hepatitis E virus in SOT patients under immunosuppressive treatment has been a concern. Therefore, mentioning or inclusion of such cases in this study would be necessary.
  2. In view of this data (Table 3), chronic Viral hepatitis caused by HBV, HCV and HEV need to be mentioned in Introduction as well as Methodology sections.

Thank you for your important comment. I have added the data regarding previous hepatitis B and C infections in the methodology section as well as in table 3.

Unfortunately, we did not collect data on hepatitis E serology during the pre-transplant period.  We therefore cannot provide this information, unless re-submitting our research project to the ethics committee.

Round 2

Reviewer 2 Report

Comments and Suggestions for Authors

The revised version looks good and satisfactory.